# Parasitological and Biochemical Efficacy of the Active Ingredients of *Allium sativum* and *Curcuma longa* in *Schistosoma mansoni* Infected Mice

**DOI:** 10.3390/molecules26154542

**Published:** 2021-07-27

**Authors:** Ali Hussein Abu Almaaty, Hadeer Abd El-hak Rashed, Maha Farid Mohamed Soliman, Eman Fayad, Fayez Althobaiti, Nahla Soliman El-Shenawy

**Affiliations:** 1Zoology Department, Faculty of Science, Port Said University, Port Said 21568, Egypt; ali_zoology_2010@yahoo.com (A.H.A.A.); hader_abdelhak@hotmail.com (H.A.R.); 2Zoology Department, Faculty of Science, Suez Canal University, Ismailia 41522, Egypt; maha_soliman@hotmail.com; 3Biotechnology Department, Faculty of Science, Taif University, P.O. Box 11099, Taif 21944, Saudi Arabia; Hhhh_fayed@yahoo.co.uk (E.F.); faiz@tu.edu.sa (F.A.)

**Keywords:** antischistosomal, praziquantel, allicin, curcumin, hematological antioxidants, DNA fragmentation

## Abstract

The active ingredients allicin and curcumin have a wide range of actions against fungi, bacteria, and helminths. Therefore, the study was aimed to evaluate the efficacy of allicin (AL) and curcumin (CU) as antischistosomal drugs and their biochemical effects in normal and *Schistosoma mansoni*-infected mice. Praziquantel (PZQ) was administrated for two successive days while AL or CU was given for two weeks from the week 7th postinfection (PI). The possible effect of different regimens on *Schistosoma* worms was evaluated by measuring the percentage of the recovered worms, tissue egg load, and oogram pattern. Serum alanine transaminase activity and levels of triglycerides, cholesterol, and uric acid were measured. Liver tissue malondialdehyde and reduced glutathione levels besides, the activities of glutathione-S-transferase, superoxide dismutase and catalase were assessed for the oxidative/antioxidant condition. DNA electrophoresis of liver tissue was used to indicate the degree of fragmentation. There was a significant reduction in the recovered worms and egg load, with a marked change of oogram pattern in all treated groups with PZQ, AL, and CU in comparison with infected-untreated mice. PZQ, AL, and CU prevented most of the hematological and biochemical disorders, as well as significantly improved the antioxidant capacity and enhanced DNA fragmentation in the liver tissue of schistosomiasis mice compared to the infected-untreated group. These promising results suggest that AL and CU are efficient as antischistosomal drugs, and it would be beneficial to test their combination to understand the mechanism of action and the proper period of treatment leading to the best result.

## 1. Introduction

Multiple Schistosoma species are parasitic to humans; *S. mansoni* and *Schistosoma haematobium* are mainly endemic to both Africa and the Middle East, which represents around 85% of the reported world cases [1]. Schistosomiasis pathogenicity can cause acute and chronic clinical syndromes [2].

Schistosomiasis caused changes in different hematological parameters [3,4]. In addition to anemia, a marked increase in monocytes, lymphocytes, and neutrophils and mild eosinophilia were recorded in Schistosomiasis [5]. Moreover, *S. mansoni* caused an elevation in the serum lipid profile and liver functions [6,7].

The eggs are first deposited by adult female *S. mansoni* in the infected host’s vasculature; then, it begins to induce a granulomatous inflammatory reaction [8]. Thus, it consequently results in pathological disorders in these tissues. *S. mansoni* caused an elevation in the levels of lipid peroxide, which reflects the increase in oxidative stress [9]. In contrast with lipid peroxide levels, authors reported that infected animals with *S. mansoni* showed a marked diminishment in different antioxidant parameters (such as catalase, reduced glutathione, and superoxide dismutase) [10,11].

Madbouly et al. [12] identified schistosomes to be genotoxic (i.e., having the capability to disrupt the genetic constitution of their hosts). The high damage observed in the schistosome-infected liver DNA is hypothesized to be caused by free radicals produced during the inflammatory response triggered by schistosomal eggs [12].

Authors recorded that the efficacy of praziquantel (PZQ) (the drug of choice) had been reduced as antischistosomal [13,14]. Scientists are still searching for antiparasitic drugs from natural origin for the development of new medications [15,16].

Allicin (AL) (the main active ingredient of *Allium sativum*) is considered to be the most critical alkaloid that is generally presumed to be responsible for their useful characteristics [17]. Most of the studies dealt with garlic itself or its aqueous or oil extracts as a medication of *Schistosomiasis mansoni* [18,19], and they were efficient in the elimination of worms and enhancement of blood indices and biochemical parameters. Serially, AL significantly caused a reduction in the worm burden, serum concentrations of liver fibrosis markers, and proinflammatory cytokines [20].

Curcumin (CU), the active ingredient of *Curcuma longa* is a naturally basic phenolic compound produced as a yellow pigment from turmeric, which is generally used as a spice and food colorant [21]. Recently, it was reported that *C. longa* extract had an antischistosomal effect, which was proven by histological, physiological and molecular studies [22].

Consequently, the present work was intended to evaluate the antischistosomal activity of the active ingredients of two essential traditional plants in infected mice and compare the findings with PZQ efficacy. Parasitological, biochemical and molecular parameters were used to qualify the efficiency of Al and CU, as well as grasp the link between these different parameters.

## 2. Results

Table 1 shows the averages and reduction rates of *S. mansoni* male and female couples and total worms recovered from all infected groups. Only a couple of worms isolated from the AL-treated group had no significant difference as compared with those of infected-untreated mice. Reduction rates of total worms recovered from PZQ, AL, and CU, were 86.0%, 73.4%, and 86.0%, respectively.

A significant reduction was recorded in the number of eggs/g liver of all three groups (3319 ± 320.4, 3813.3 ± 494.4, 1937.6 ± 543.4 for PZQ, AL, and CU-treated mice, respectively), as compared with an infected-untreated group (6513.4 ± 1127.6) (Table 2). A significant reduction was recorded in the number of eggs/g intestine of PZQ and CU treated-groups only when compared with infected untreated mice (Table 2).

The three treatment regimens significantly changed the oogram patterns when compared with the infected-untreated group in both the liver and intestine (Table 3). Regarding the oogram pattern in the liver, the highest percentage of dead eggs was recorded for CU-treated mice (43%; *p* < 0.002), as compared to infected-untreated animals (1.60%). In the case of the intestine, the highest percentage of dead eggs was recorded for the PZQ-treated group (41.67%; *p* < 0.001) in comparison with infected-untreated mice (0.8%).

Table 4 shows the changes in the averages of erythrocytes and blood indices of mice. In uninfected animals, there were not any significant changes in the average of the number of RBCs in mice treated with PZQ, AL, or CU as compared to their control groups, while there was a significant diminishment in the HGB average in the group treated with PZQ or AL in comparison with saline-treated mice.

The study revealed that *S. mansoni* caused a significant reduction in the averages of RBCs number and HGB (4.4 ± 0.2 and 8.2 ± 0.4), respectively, in infected untreated mice when compared with the control saline group. The animals treated with PZQ, AL, or CU showed a significant increase in the HGB average when compared to the infected-untreated group. Other blood indices average HCT, MCV, MCH, MCHC, RDW-SD, and RDW-SD were in turn influenced by RBCs and HGB (Table 4).

Table 5 shows changes in the averages of leucocytes of different groups. In uninfected groups, WBCs averages in mice treated with PZQ were significantly reduced to 3.40 ± 0.23 and were 5.67 ± 0.15 for the AL-group in comparison with saline treated-mice (4.30 ± 0.12). However, there was not any significant change in the WBCs of the CU group as compared with the PB-group. Concerning the infected groups, there was a considerable increase in the averages of WBCs by 5.76-fold when compared with the saline group. The average of the WBCs in the groups treated with PZQ, AL, or CU diminished by 32.8%, 58.26%, and 53.25%, respectively. The averages of LYM, MID, and GRAN in all groups are shown in Table 5.

The changes in the averages of platelets and their indices of mice were presented in Table 6. In uninfected groups, the only significant difference was recorded for P-LCR in the CU-treated group as compared with PB mice. In the infected untreated group, significant differences were recorded for MPV and PCT when compared with the saline group. There was a significant increase in MPV and PCT averages in all treated groups as compared with infected untreated animals. PZQ-treated mice showed a remarkable change in PDW. In AL and CU animals, the averages of P-LCR were significantly increased by 2.8- and 1.1-fold, respectively, in comparison with the infected-untreated group.

The averages of serum TC and TG levels, as well as the ALT activity of different groups, were shown in Figure 1A,B, respectively. In uninfected mice, the AL-treated group showed a significant reduction in the average of TC level as compared with saline-treated mice. TC level and ALT activity averages in mice treated with CU were significantly decreased by 15.1% and 17.8% when compared with the PB-group, respectively. In infected untreated mice, there was a significant increase in the averages TC and TG levels (as well as ALT activity) as compared with the regular saline group. All infected treated groups showed significant diminishment in the average levels of TC, TG, and the activity of ALT when compared with infected untreated ones.

AL caused significant diminish in UA levels in uninfected and infected mice by 22.9% and 27.3% (Figure 2) when compared with saline and infected untreated mice, respectively. However, there was a different significant change in PZQ uninfected and infected treated mice by 1.3- and 2.4-fold, respectively.

The changes in the averages of MDA levels of mice in different groups were presented in Figure 3. Uninfected groups treated with AL or CU had a significant diminishment in the levels of MDA when compared with their controls. Infected groups treated with PZQ or AL or CU showed a marked decrease in the averages to be 104.0 ± 2.6, 99.7 ± 6.7 and 95.2 ± 6.2, respectively, as compared with infected untreated mice (147.1 ± 6.3).

Figure 4 shows the changes in the averages of the antioxidant parameters of mice. In uninfected animals, there was a significant increase in GST activity for the AL-group when compared with saline-treated mice and for mice treated with CU as compared with the PB-group (Figure 4A). Regarding the infected treated groups with PZQ, AL, or CU, there was a significant increase in the averages of all measured antioxidant parameters (GSH, GST, SOD, and CAT) when compared with those of infected-untreated mice (Figure 4A,B). The averages of GSH, GST, SOD, and CAT recorded the highest significant increase in CU-treated animals when compared with infected untreated groups.

Results obtained through the gel electrophoresis of liver tissues are shown in Figure 5. No specific fragmentation was observed in the liver tissues of the saline control group (Lane 1). On the contrary, there was a marked laddering fragmentation in the infected-untreated groups (Lane 2). There was a restoration of DNA pattern isolated from liver tissues on infected groups treated with the three different materials (PZQ, AL, and CU), as shown in Lanes 3, 4 and 5, respectively.

## 3. Discussion

The present work is one of the few studies dealing with the efficacy of the active ingredients AL and CU compared with PZQ in the treatment of schistosomiasis using different aspects (parasitological, biochemical, and DNA fragmentation).

The reduction rates of the worms were convergent in the PZQ and CU groups (86.02% and 86%, respectively). On the contrary, with the present result, *C. longa* extract alone showed a low reduction rate of the worms that didn’t exceed 35% [22]. This observation may improve the outbalance efficacy of the active ingredient of *C. longa* more than that of the plant itself.

The highest reduction rate of egg load in liver tissues was recorded for CU-treated mice (70.25%), while PZQ-treated animals showed the highest reduction rate of egg load/intestine to be 91.94%. The sequence of dead egg percentage from the highest to the lowest amongst PZQ, AL, or CU groups in the liver and intestine was similar to the sequence of egg load reduction rates. The lowest reduction rates of AL than PZQ or CU are in agreement with Metwally et al. [20], who recorded that AL caused the lowest reduction rate between PZQ and garlic in either worm burden or egg load.

There was a significant reduction in RBC’s count, HGB, HCT, MCV, and MCHC in infected-untreated mice as compared with the health saline group that is in parallel with recent research [4]. The reduction in RBC’s count, HGB, and HCT parameters reflected severe anemia that may be due to blood loss and improved rate of hemolysis as well as the shortened life span of RBCs. Blood may be lost either from the bleeding induced by the disposal of the egg through the intestinal wall or due to consumption by adult schistosomes [23]. AL had a more significant effect on RBCs and HCT of the infected mice than PZQ and CU, which may be due to a reduction of worms and egg load. Consistently with this result, it was reported previously that *S. mansoni* infected groups treated with garlic oil or aqueous garlic extracts (or a mixture of both) showed a significant improvement in HGB and HCT in comparison with infected-untreated animals [19]. Also, CU-infected treated mice showed an increase in the values of RBCs and HCT after two weeks of treatment [24,25].

PZQ, Al and CU caused a significant reduction in all leukocyte parameters of the infected groups as compared with infected untreated animals. This decrease in leukocytosis in Al infected-treated mice agreed with aqueous garlic extract, which caused a reduction in WBC’s average [18]. However, the efficacy of CU could be due to its ability to stabilize the cell membrane and restore variable hematological indices [26].

Most of the previous studies focused on the relationship between antischistosomal drugs and the PLT count. Alkazzaz et al. [27] reported that PLT counts generally decreased in *S. mansoni* infected mice and the degree of thrombocytopenia increased in response to the time of infection. In the present result, there was not any significant difference in PLT among different groups, either infected or not.

The irritation of hepatocytes produced by the emitted toxins or metabolic products of the growing young worm, adult worms, and eggs can result in liver damage [28], which is associated with the increase of some enzyme activities in serum as ALT [29]. This illustrates the significant rise of ALT activity by 1.2-folds in infected-untreated mice in comparison with the control group. PZQ, AL, and CU caused a reduction in ALT activities in infected groups. This was in collaboration with the efficacy of the drugs to eliminate worms and eggs, which in turn caused the enzymatic enhancement of the hepatocytes.

The highest reduction levels of TC and TG among the infected treated-groups, in comparison with infected untreated ones, was recorded for the CU drug to be 16.9% and 19.5%, respectively. One hypothesis is that CU prevents the rise in serum cholesterol in animal studies by inhibiting dietary cholesterol absorption [30].

The present work showed that *S. mansoni* infection caused a significant elevation in the hepatic MDA levels, which is the representative of oxidative stress and marked depletion in the antioxidants; GSH, GST, SOD, and CAT. Infected groups treated with PZQ, AL, or CU showed improvement in these parameters in comparison with infected-untreated mice. Garlic and its derivatives (as well as CU) were investigated previously as antischistosomal drugs, and they were also found to act as proper antioxidants [10,19,24,31].

The antioxidant mechanism of AL in the scavenging of free radicals was shown to be able to scavenge O^2·^ and OH· [32]. However, CU was recorded as a potent antioxidant in vitro and in vivo due to its ability to slow down the production of pro-oxidants such as ROS and GSSG [33,34].

In addition, the molecular applications were used in the evaluation of the therapies’ efficacy against *S. mansoni* infection [35]. DNA fragmentation and the diminishment of antioxidants are correlated with severe hepatotoxicity and histological changes [36]. In the present work, the laddering DNA is considered to be a reflection of fragmentation, and this is integrated with the pathological disorders recorded for oxidative stress and antioxidants in infected-untreated mice as mentioned above.

*S. mansoni* infection caused genetic alterations in the DNA of mice liver as previously reported by Riad et al. [37]. The PZQ, AL, and CU caused marked enhancement in the DNA pattern of infected liver tissues. AL was found to improve such deformations to a great extent. This was illustrated by the enhancement of DNA fragmentation in the liver of infected treated-mice with the active ingredient of garlic. The data of PZQ disagrees with Eid et al. [38], who found that infected animals treated with PZQ showed no significant change in DNA fragmentation pattern in comparison with the infected-untreated group. However, the combination of PZQ and CU showed a marked enhancement of damaged DNA in infected animals [22].

## 4. Material and Methods

### 4.1. Chemicals

Allicin (AL) (C_6_H_10_OS_2_ & ≥ 80% HPLC) was obtained from Science Med. (Egypt) and dissolved in distilled water. Curcumin (CU) (C_21_H_20_O_6_ & ≥ 65% HPLC) was brought from Sigma Aldrich, mixed with 0.5 M NaOH, and then suspended in a phosphate buffer (PB) to obtain the desired concentration according to vial instructions. PZQ was obtained from the Egyptian International Pharmaceutical Industries Co (E.I.P.I.CO.). All of the commercial kits for biochemical parameters were bought from the Spin-react Company (Spain), while oxidative stress and antioxidant kits were purchased from the Bio-diagnostic Company (Egypt).

### 4.2. Experimental Animals and Design

The fewest number of mice were chosen for this research to have valid results statistically. Ninety CD1 male, white albino mice with an average body weight of 18 ± 2.6 g were purchased from Theodore Bilharz Research Institute (TBRI, Imbaba, Giza, Egypt). They were kept in plastic cages (ten mice/cage) and held in the animal house of the Zoology Department, Faculty of Science, Port-Said University. They were placed under standard conditions maintained at a room temperature (20–25 °C), exposed to a 12 hr light/dark cycle and had free access to pellet food with tap water ad libitum. Mice were divided in a random manner into nine groups (10 animals/group) according to the experimental design shown in Table 7. Four groups were injected subcutaneously with 60 ±10 *Schistosoma mansoni* cercariae (Egyptian strain) per animal. Two groups of PZQ animals were treated orally (PO) from the 7th week for two successive days, while the other groups were injected intraperitoneally (IP) with different regimens from the 7th week of infection for two weeks (three times per week).

The dose of treatment in PZQ treated-groups (noninfected and infected) was selected according to Chaiworaporn et al. [39]. Moreover, in the preliminary study on the effect of different doses of AL (10, 20, and 40 mL/kg body weights of mice) and CU (10, 20, and 40 mg/kg body weights of mice), we found the highest dose was the most effective in the elimination of helminthes and, thus, was used in this experimental animal design for further tests.

Animals in all groups (healthy and infected) described as fasted were deprived of food for 12 h but allowed free access to tap water after 14 days from the beginning of treatment and before collecting the blood and organ samples. By the end of the experimental period, the dissection and taking of the blood were performed under anesthesia to avoid any stress and pain that this process could inflict on the mice.

### 4.3. Collection of Blood and Organs

Blood samples of the fasted mice were collected immediately from the medial retro-orbital venous plexus using capillary tubes (Micro Hematocrit Capillaries, Mucaps) under ether anesthesia [40]. Then, the blood was centrifuged at 3000 rpm for 15 min and collected serum samples were marked and stored at −20 °C until they were used for different biochemical tests. Some blood samples were collected in a clean EDTA tube for the determination of complete blood count (CBC) [41].

From each group, parts of the liver and intestine were removed from some mice under ether anesthesia for parasitological assays. However, other parts of the liver tissues in the same group were either perfused with cold buffer containing 1.15% of KCl and 0.5 mM of EDTA, blotted dry on filter paper and stored at −80 °C for further biochemical analyses in tissues or stored directly at −80 °C for molecular analysis.

### 4.4. Worm Recovery

Each liver was put at once into a plastic folder and compressed between glass plates until the parenchyma was evenly dispersed into a delicate transparent layer and examined under a stereomicroscope to count the worms and classify them into male (♂), female (♀), and copulated [42].

The small and large intestines were removed and located in a Petri dish to examine the mesenteric veins under a stereoscopic microscope [43]. All *S. mansoni* worms were removed, counted, and classified. The reduction in the recovered worms from treated mice compared with untreated ones was expressed in a percentage formula as follows: P = (C − V)/C × 100 [44], where P indicates the percentage of the reduction, C indicates the number of worms isolated from the infected groups, and V indicates the number of parasites recovered from treated groups.

### 4.5. Ova Count

0.5 g from isolated livers and intestines were put individually in a falcon tube containing 5% KOH solution and placed in the incubator at 37 °C for 24 h for the complete digestion of tissues [45]. The total egg count was expressed as the mean number of eggs/mg of liver and intestine [46]. The percentage of reduction was calculated according to the equation: P = (C − V)/C × 100 [47].

### 4.6. Oogram

Three fragments of the liver and small intestine were cut longitudinally, rinsed in saline, softly dried on filter paper, and then compressed between two glass slides to obtain the thin preparation. Slides were examined under 10× microscope power and the stage of each egg was recorded in each fragment [48]. One hundred eggs were counted in each piece and classified according to their developmental stage [49].

### 4.7. Biochemical Serum Parameters

The level of total cholesterol (TC) and triglycerides (TG) were estimated according to the methods of Naito [50] and Buccolo and David [51], respectively. The activity of alanine transaminase (ALT) was assessed according to Murray [52]. The serum uric acid (UA) level was determined according to Schultz [53].

### 4.8. Determination of Oxidative Stress/Antioxidant Parameters in Liver Tissue

Levels of lipid peroxide in the liver tissues were estimated according to Ohkawa et al. [54]. The level of reduced glutathione (GSH) was estimated according to Beutler et al. [55]. Glutathione-s-transferase (GST) activity was determined using the method of Habig et al. [56]. Superoxide dismutase (SOD) and Catalase (CAT) were evaluated according to Nishikim et al. [57] and Aebi [58], respectively.

### 4.9. Detection of DNA Fragmentation by Agarose Gel Analysis

Before the DNA extraction process, liver tissues from infected mice were perfused to remove any attached worms, to be sure that the fragmentation result was a reflection of the liver condition only. The liver tissues of the infected and control saline groups were subject to DNA extraction after 24 hrs according to Sambrook and Russel [59]. The concentration of DNA was analyzed via electrophoresis on 2% agarose gel containing 1% GelRed (1:500) (Biotium, Hayward, EUA).

### 4.10. Statistical Analysis

Data were given as mean ± standard error (Mean ±SE). All data were analyzed using the Statistic Program Sigma Stat (SPSS), version 20. The effects of different regimens was analyzed via one-way ANOVA (Analysis of variance). A value of *p* < 0.05 was interpreted as statistically significant [60].

## 5. Conclusions

There is an obvious correlated relationship between *Schistosoma mansoni* infection and some pathological disorders (such as hematological, oxidative stress, and antioxidant changes), which can extend to DNA malformations. This promising result showed the proper efficacy of AL and CU as antischistosomal drugs which were close to PZQ in the elimination of adult worms and eggs and, at the same time, the improvement of MDA and antioxidants caused an enhancement of the DNA fragmentation pattern. In addition, there is a need to test the combination of these three drugs as antischistosomiasis, a greater understanding the mechanism of action of AL and CU, and the more preop period of treatment leading to the best result.

## Figures and Tables

**Figure 1 molecules-26-04542-f001:**
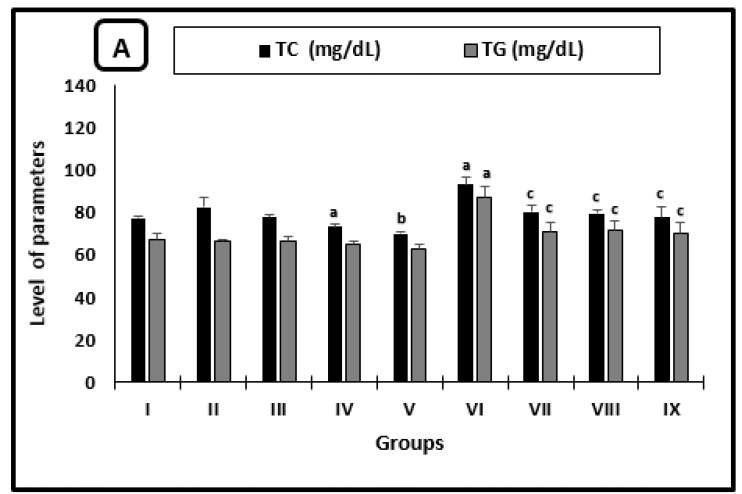
Serum parameters changes in the mice after 2 weeks of treatment. (**A**) Cholesterol and triglycerides levels, (**B**) ALT activity. ^a^ significant difference from the saline group, ^b^ significant difference from the phosphate buffer group, ^c^ significant difference from the infected-untreated group. Groups I, II, III, IV, and VI presented saline, PB, PZQ, AL and CU uninfected treated-mice, groups VI, VII, VIII and IX presented infected-untreated, PZQ, AL and CU treated mice.

**Figure 2 molecules-26-04542-f002:**
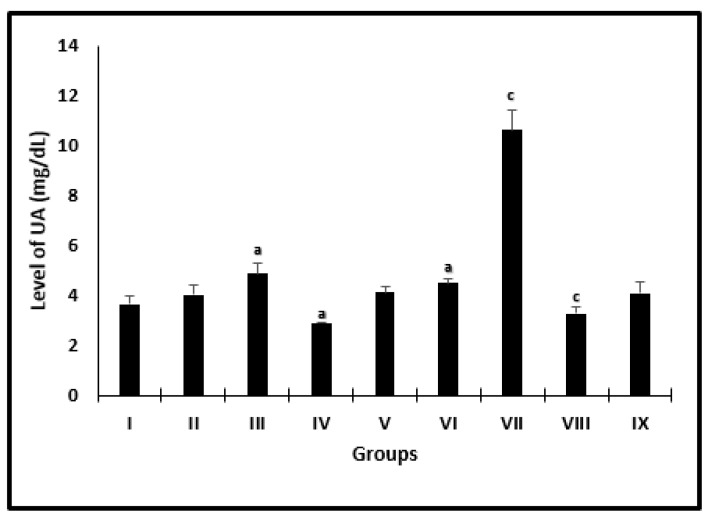
Serum uric acid level changes in mice after 2 weeks of treatment. ^a^ significant difference from the saline group, ^b^ significant difference from the phosphate buffer group, ^c^ significant difference from the infected untreated group. Groups I, II, III, IV, and VI presented saline, PB, PZQ, AL and CU uninfected treated-mice, and groups VI, VII, VIII, and IX presented infected-untreated, PZQ, AL, and CU treated-mice.

**Figure 3 molecules-26-04542-f003:**
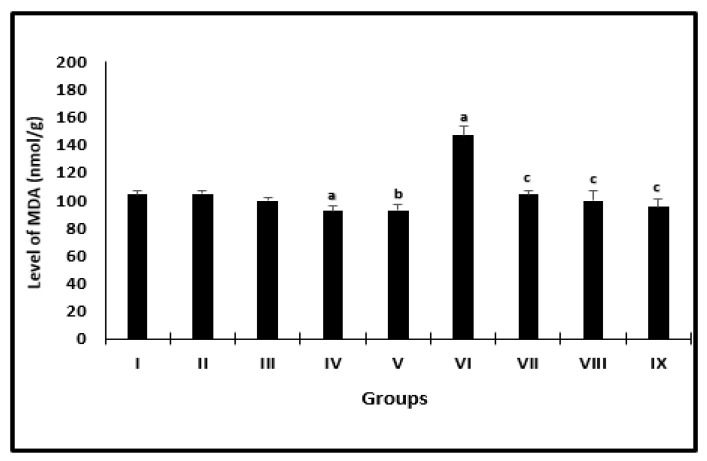
Liver MDA level of mice after 2 weeks of treatment. ^a^ significant difference from the saline group, ^b^ significant difference from the phosphate buffer group, ^c^ significant difference from the infected-untreated group. Groups I, II, III, IV, and VI presented saline, PB, PZQ, AL, and CU uninfected treated mice, groups VI, VII, VIII, and IX presented infected-untreated, PZQ, AL, and CU presented treated mice.

**Figure 4 molecules-26-04542-f004:**
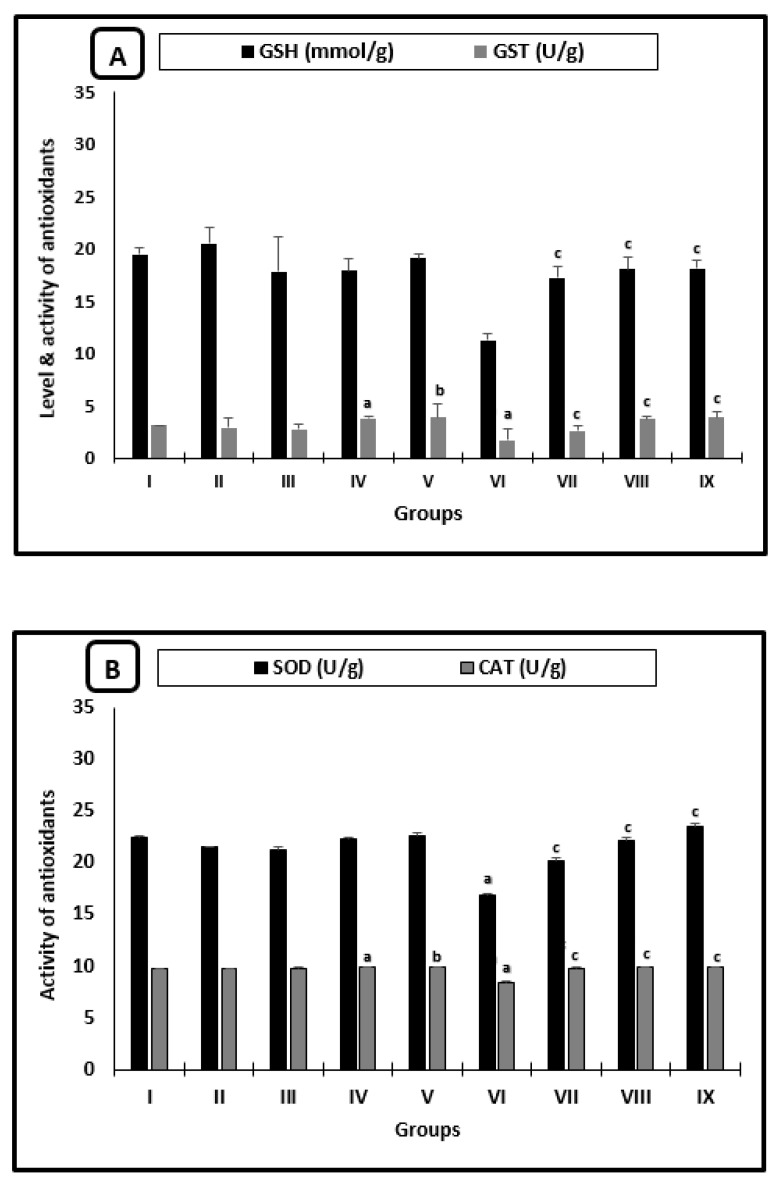
Liver antioxidant changes of mice after 2 weeks of treatment. (**A**) GSH levels and GST activity. (**B**) SOD and CAT activities. ^a^ significant difference from the saline group, ^b^ significant difference from the phosphate buffer group, ^c^ significant difference from the infected-untreated group. Groups I, II, III, IV, and VI presented saline, PB, PZQ, AL, and CU uninfected treated mice, groups VI, VII, VIII, and IX presented infected-untreated mice, and PZQ, AL, and CU presented treated mice.

**Figure 5 molecules-26-04542-f005:**
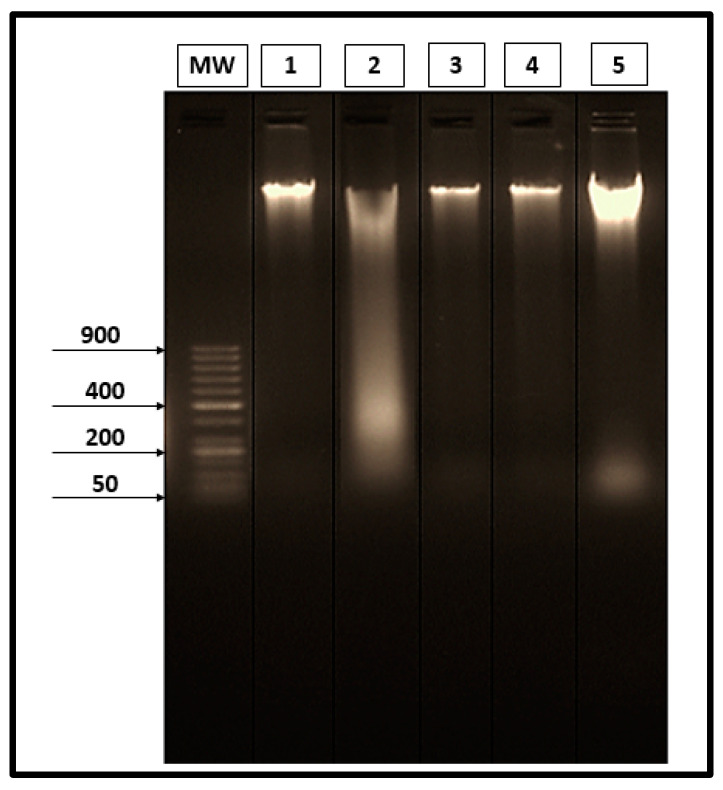
Electrophoretic separated genomic DNA from liver tissues. (Lane MW) 50 bp DNA ladder marker. (Lane 1) DNA extracted from the liver of the saline group. (Lane 2) DNA extracted from the liver of the infected-untreated group. (Lane 3) DNA extracted from the liver of the infected group treated with PZQ. (Lane 4) DNA extracted from the liver of the infected group treated with AL. (Lane 5) DNA extracted from the liver of the infected group treated with CU.

**Table 1 molecules-26-04542-t001:** Worm load and reduction rate of *S. mansoni* infected mice after different treatment regimens (8 weeks pi).

Groups	Worm Burden							
	Male	% Reduction	Female	% Reduction	Couples	% Reduction	Total	% Reduction
VI	16.8 ± 2.93	-	19.8 ± 1.84	-	10.40 ± 2.05	-	47 ± 5.15	-
VII	2.57 ± 0.8 ^c^	84.7	3.43 ± 0.48 ^c^	82.68	0.57 ± 0.30 ^c^	94.52	6.57 ± 1.07 ^c^	86.02
VIII	3.13 ± 1.2 ^c^	81.37	3.13 ± 0.44 ^c^	84.19	6.25 ± 1.26	39.90	12.51 ± 2.11 ^c^	73.38
IX	2.29 ± 0.6 ^c^	86.37	2.29 ± 0.42 ^c^	88.43	2 ± 0.72 ^c^	80.77	6.58 ± 1.36 ^c^	86

Data presented as mean ± S.E, ^c^ significant difference from the infected untreated group. Groups VI, VII, VIII and IX presented infected-untreated, PZQ, AL, and CU treated-mice.

**Table 2 molecules-26-04542-t002:** Egg load per gram of liver and intestine of *S. mansoni* infected mice and reduction rate after different treatment regimens (8 weeks pi).

Groups	Mean Egg Load/Gram of Liver		Mean Egg Load/Gram of Intestine	
	Mean	% Reduction	Mean	% Reduction
VI	6513.4 ± 1127.59	0	18906.6 ± 1362.64	0
VII	3319 ± 320.38 ^c^	49.04	1523.86 ± 120.97 ^c^	91.94
VIII	3813.25 ± 494.39 ^c^	41.46	4189.25 ± 117.12	77.84
IX	1937.57 ± 543.37 ^c^	70.25	2992.86 ± 681.98 ^c^	84.17

Data presented as mean ± S.E, ^c^ significant difference from the infected untreated group. Groups VI, VII, VIII, and IX presented infected-untreated, PZQ, AL, and CU treated-mice.

**Table 3 molecules-26-04542-t003:** Oogram pattern showing the percentage of *S. mansoni* ova at different stages of maturity in the liver and the intestine of infected mice after different treatment regimens (8 weeks pi).

Groups		% Egg Developmental Stages of Liver					% Egg Developmental Stages of Intestine			
	Early Immature	Late Immature	Total Immature	Mature	Dead	Early Immature	Late Immature	Total Immature	Mature	Dead
VI	17.00	22.40	39.40	59.00	1.60	14.60	32.40	47.00	52.20	0.80
VII	8.50	6.67 ^c^	15.17 ^c^	42.50	42.33 ^c^	4.17 ^c^	7.50 ^c^	11.67 ^c^	46.66	41.67 ^c^
VIII	4.63 ^c^	13.25 ^c^	17.88 ^c^	44	38.12 ^c^	8.50 ^c^	31.88	40.38	22.12 ^c^	37.50 ^c^
IX	4.71 ^c^	6.58 ^c^	11.29 ^c^	45.71	43 ^c^	2.29 ^c^	11.14 ^c^	13.43 ^c^	47.43	39.14 ^c^

Data presented as mean ± S.E, ^c^ significant difference from the infected untreated group. Groups VI, VII, VIII, and IX presented infected-untreated, PZQ, AL, and CU treated-mice.

**Table 4 molecules-26-04542-t004:** Changes in erythrocytes and blood indices of mice after 2 weeks of the treatment.

Group	RBC (×10^6^ µL)	HGB (g/dl)	HCT (%)	MCV (Fl)	MCH (pg)	MCHC (g/Dl)	RDW-SD (fL)	RDW-CV (%)
I	6.49 ± 0.25	12.50 ± 0.17	34.35 ± 0.66	53.10 ± 0.77	19.32 ± 0.84	36.40 ± 0.20	29.83 ± 0.07	14.70 ± 0.06
II	5.70 ± 0.05	10.40 ± 0.40	28.25 ± 0.38	49.59 ± 0.64	18.25 ± 0.55	36.82 ± 1.45	33.10 ± 0.64	15.80 ± 0.06
III	5.72 ± 0.13	10.75 ± 0.10 ^a^	28.40 ± 0.12 ^a^	49.67 ± 0.97	18.79 ± 0.31	37.84 ± 0.35 ^a^	27.77 ± 0.03 ^a^	13.65 ± 0.03 ^a^
IV	6.65 ± 0.27	11.23 ± 0.32 ^a^	30.60 ± 0.91 ^a^	46.18 ± 2.20 ^a^	16.96 ± 0.85	36.71 ± 0.13	27.77 ± 0.03 ^a^	13.70 ± 0.12 ^a^
V	6.39 ± 0.25	10.60 ± 0.35	29.45 ± 0.20 ^b^	46.20 ± 1.90	16.64 ± 0.88	35.98 ± 0.93	27.77 ± 0.03 ^b^	13.65 ± 0.03 ^b^
VI	4.35 ± 0.18 ^a^	8.17 ± 0.38 ^a^	21.2 ± 0.95 ^a^	48.58 ± 1.43 ^a^	18.76 ± 0.40	38.70 ± 0.42 ^a^	32.26 ± 0.16 ^a^	15.95 ± 0.26 ^a^
VII	5.27 ± 0.20 ^c^	10 ± 0.17 ^c^	26.22 ± 0.65 ^c^	53.28 ± 5.14	20.31 ± 1.89	38.17 ± 1.09	29.38 ± 0.07 ^c^	14.30 ± 0.12 ^c^
VIII	5.70 ± 0.16 ^c^	9.63 ± 0.20 ^c^	26.60 ± 0.60 ^c^	46.63 ± 0.90	16.90 ± 0.15 ^c^	36.26 ± 0.14 ^c^	31.67 ± 0.33	14.83 ± 0.09 ^c^
IX	5.56 ± 0.14 ^c^	9.87 ± 0.38 ^c^	25.27 ± 0.18 ^c^	45.54 ± 1.35	17.75 ± 1.07	39.05 ± 1.52	29.20 ± 1.40	14.53 ± 0.43 ^c^

Data presented as mean ± S.E, (N = 5–8). ^a^ significant difference from the saline group, ^b^ significant difference from the phosphate buffer group, ^c^ significant difference from the infected untreated group. Groups I, II, III, IV, and VI presented saline, PB, PZQ, AL, and CU uninfected treated-mice, groups VI, VII, VIII, and IX presented infected-untreated, PZQ, AL, and CU treated-mice. RBC; erythrocytes number, HGB; hemoglobin concentration, HCT; hematocrit, MCV; mean corpuscular volume is the average volume of red cells in a specimen, MCH; The mean corpuscular hemoglobin (MCH), (mean cell hemoglobin is the average mass of hemoglobin per red blood cell in a sample of blood), MCHC; mean corpuscular hemoglobin concentration is the average concentration of hemoglobin in your red blood cells. RDW-SD; red blood cell distribution width, RDW-CV; is elevated in accordance with variation in red cell size.

**Table 5 molecules-26-04542-t005:** Changes in leucocytes of mice after 2 weeks of the treatment.

Group	WBC (10^3^ µL)	LYM (10^3^ µL)	MID (10^3^ µL)	GRAN (10^3^ µL)
I	4.30 ± 0.12	3.90 ± 0.29	0.30 ± 0.06	0.10
II	5.40 ± 0.70	3.25 ± 0.25	1.35 ± 0.29	0.80 ± 0.10
III	3.40 ± 0.23 ^a^	2.50 ± 0.17 ^a^	0.65 ± 0.09 ^a^	0.25 ± 0.02 ^a^
IV	5.67 ± 0.15 ^a^	4.67 ± 0.19	0.83 ± 0.02 ^a^	0.23 ± 0.02 ^a^
V	6.15 ± 0.66	4.70 ± 0.29 ^b^	0.85 ± 0.14	0.60 ± 0.23
VI	24.75 ± 2.69 ^a^	19.02 ± 0.66 ^a^	3.88 ± 0.04 ^a^	1.86 ± 0.32 ^a^
VII	16.63 ± 0.90 ^c^	11.43 ± 0.07 ^c^	3.35 ± 0.07 ^c^	1.85 ± 0.03
VIII	10.33 ± 0.75 ^c^	8.28 ± 0.52 ^c^	1.22 ± 0.13 ^c^	0.83 ± 0.06
IX	11.57 ± 1.29 ^c^	9.55 ± 0.31 ^c^	1.42 ± 0.18 ^c^	0.60 ± 0.06 ^c^

Data presented as mean ± S.E, (N = 5–8). ^a^ significant difference from the saline group, ^b^ significant difference from the phosphate buffer group, ^c^ significant difference from the infected untreated group. Groups I, II, III, IV, and VI presented saline, PB, PZQ, AL, and CU uninfected treated mice, groups VI, VII, VIII, and IX presented infected-untreated, PZQ, AL, and CU treated mice. WBC; leukocyte number, LYM; lymphocytes, MID; midrange absolute (monocytes, eosinophil, and basophil), GRAN; granulated cells (neutrophils).

**Table 6 molecules-26-04542-t006:** Changes in platelets and their indices of mice after two weeks of the treatment.

Group	PLT (×10^3^ µL)	MPV (fL)	PDW (%)	PCT (%)	P-LCR (%)
I	409.50 ± 17.03	6.77 ± 0.20	7.87 ± 0.03	0.27 ± 0.01	0
II	461.50 ± 2.60	7.10 ± 0.12	8.15 ± 0.14	0.31	0
III	376.17 ± 14.01	6.45 ± 0.9	7.87 ± 0.03	0.23 ± 0.03	0
IV	394.67 ± 18.75	6.63 ± 0.13	7.87 ± 0.03	0.26 ± 0.01	0
V	480.28 ± 15.16	7.10 ± 0.46	8.35 ± 0.26	0.30 ± 0.08	3 ± 0.29 ^b^
VI	375.83 ± 12.52	6.11 ± 0.10 ^a^	8.15 ± 0.14	0.21 ± 0.02 ^a^	0
VII	388.50 ± 10.10	6.95 ± 0.14 ^c^	8.82 ± 0.16 ^c^	0.27 ± 0.01 ^c^	0
VIII	387.33 ± 14.43	7.03 ± 0.18 ^c^	8.70 ± 0.68	0.27 ± 0.01 ^c^	2.76 ± 0.33 ^c^
IX	395.67 ± 82.49	6.39. ± 0.03 ^c^	8.67 ± 0.44	0.28 ± 0.01 ^c^	1.13 ± 0.13 ^c^

Data presented as mean ± S.E, (N = 5–8). ^a^ significant difference from the saline group, ^b^ significant difference from the phosphate buffer group, ^c^ significant difference from the infected untreated group. Groups I, II, III, IV, and VI presented saline, PB, PZQ, AL, and CU uninfected treated-mice, groups VI, VII, VIII, and IX presented infected-untreated, PZQ, AL, and CU treated-mice. PLT; platelets, MPV; mean platelets volume, PDW; platelet distribution width, PCT, plateletcrit; mean platelet volume, platelet distribution width: its expected values and correlation with parallel red blood cell parameters, P-LCR; platelet large cell ratio.

**Table 7 molecules-26-04542-t007:** Experimental design.

Groups	Drug Regimens of Groups	Dose
Noninfected control	Group I: saline Group II: PB Group III: PZQ Group IV: AL Group V: CU	0.1 mL of 0.9% NaCl 0.1 mL of 1 M PB buffer pH 7.2 0.1 mL of PZQ (300 mg/kg body weight of mice) 0.1 mL of AL (40 mL/kg body weight of mice) 0.1 mL of CU (40 mg/kg body weight of mice)
Infected	Group VI: untreated Group VII: PZQ Group VIII: AL Group IX: CU	0.1 mL of 0.9% NaCl 0.1 mL of PZQ (300 mg/kg body weight of mice) 0.1 mL of AL (40 mL/kg body weight of mice) 0.1 mL of CU (40 mg/kg body weight of mice)

## Data Availability

The data presented in this study are available on request from the corresponding author.

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
