# Peer review of "Parasitological and Biochemical Efficacy of the Active Ingredients of Allium sativum and Curcuma longa in Schistosoma mansoni Infected Mice"

_molecules, 2021, doi:10.3390/molecules26154542_

Round 1
Reviewer 1 Report
Schistosomiasis is, next to malaria, the most serious parasitological health problem in the world.
Studies aimed at increasing the effectiveness of treatment and at the same time reducing the side effects caused by the use of drugs approved for treatment are very important. Also, studies confirming the possibility of using allicin and curcumin with praziquantel are important and may be used in the treatment in the future.
Author Response
Thanks for your comments. Also thanks for your effort and time
best regard
Nahla
Reviewer 2 Report
The methodology and the plan of the experiment were developed according. All determined indicators suggest that Allicin (AL) and Curcumin (CU) positively affect the parasite Schistosoma mansoni in mice. The issue is topical as we know that there is less and less healthy drinking water today, and therefore the fight against parasites is becoming more topical.
Schistosoma mansoni is a water-borne parasite of humans, and belongs to the group of blood flukes (Schistosoma). The adult lives in the blood vessels (mesenteric veins) near the human intestine. Clinical symptoms are caused by the eggs. As the leading cause of schistosomiasis in the world, it is the most prevalent parasite in humans. The life cycle of schistosomes includes hosts humans as definitive hosts, where the parasite undergoes sexual reproduction. S. mansoni is transmitted through water. The larvae are able to live in water and infect the hosts by directly penetrating the skin. Prevention of infection is done by improved sanitation. Infection is treated with praziquantel.
The study aimed to evaluate the efficacy of allicin (AL) and curcumin (CU) as anti-schistosomal drugs and their biochemical effects in normal and Schistosome mansoni infected mice. Praziquantel (PZQ) was given for two consecutive days, while AL or CU was given for two weeks from the seventh week after infection (PI).
- mansoni infection caused a significant increase in liver MDA levels, which is representative of oxidative stress and a marked decrease in antioxidants; GSH, GST, SOD and CAT. Infected groups treated with PZQ or AL or CU showed improvement in these parameters compared to infected untreated mice.
The results obtained using Allicina (AL) from Allum sativum and Curcumina (CU) from Curcuma longa, which are natural compounds based on the phenolic component, give promising results for the purpose of replacing the synthetic drug (PZO) praziquantel.
The obtained results are a solid basis for further research on natural remedies.
Author Response
Thanks for your comments, effort, and time. We are going to improve the English of the text and check the spell.
Best regards
Nahla
Reviewer 3 Report
In this manuscript, the authors focused on evaluating the efficacy of allicin and curcumin as anti-schistosomal drugs and their biochemical effects in normal and Schistosoma mansoni-infected mice.
The approach used for these techniques is scientifically sound and the results correctly presented. The theoretical research involved in this manuscript is of great significance. But the author did not elaborate on the important significance. It is suggested that the significance of the manuscript research should be emphasized and the rigor of the manuscript logic should be increased. However, the English in the manuscript is understandable but can be improved. The authors should revise the manuscript for language. Moreover, the authors should supplement some more meaningful experiments and refer to more literatures to show the importance of this work. Overall, I don't think the manuscript is suitable for publication.
Specific comments
- The introduction part lacks logic and the research significance of this manuscript is not highlighted. It is recommended to reorganize.
- Standard substances such as allicin and curcumin in the manuscript should be marked with purity.
- Payattention to the format of references, and there are too many references in this manuscript. Too many references can not show the level of the manuscript well. It is recommended to cite references in recent years.
- I think the results part is just a simple listing,and does not deeply link the experiment results to explain
- The presentation of theexperimen results is too simple and not very attractive.
- The definition of the Figurehorizontal and vertical coordinates is not high, it is recommended to adjust.
Author Response
Reply to the referees
Schistosomiasis is, next to malaria, the most serious parasitological health problem in the world.
Studies aimed at increasing the effectiveness of treatment and at the same time reducing the side effects caused by the use of drugs approved for treatment are very important. Also, studies confirming the possibility of using allicin and curcumin with praziquantel are important and may be used in the treatment in the future.
Thanks for your variable comments.
We are going to improve the English and check the spell
The methodology and the plan of the experiment were developed according. All determined indicators suggest that Allicin (AL) and Curcumin (CU) positively affect the parasite Schistosoma mansoni in mice. The issue is topical as we know that there is less and less healthy drinking water today, and therefore the fight against parasites is becoming more topical.
Schistosoma mansoni is a water-borne parasite of humans, and belongs to the group of blood flukes (Schistosoma). The adult lives in the blood vessels (mesenteric veins) near the human intestine. Clinical symptoms are caused by the eggs. As the leading cause of schistosomiasis in the world, it is the most prevalent parasite in humans. The life cycle of schistosomes includes hosts humans as definitive hosts, where the parasite undergoes sexual reproduction. S. mansoni is transmitted through water. The larvae are able to live in water and infect the hosts by directly penetrating the skin. Prevention of infection is done by improved sanitation. Infection is treated with praziquantel.
The study aimed to evaluate the efficacy of allicin (AL) and curcumin (CU) as anti-schistosomal drugs and their biochemical effects in normal and Schistosome mansoni infected mice. Praziquantel (PZQ) was given for two consecutive days, while AL or CU was given for two weeks from the seventh week after infection (PI).
- mansoni infection caused a significant increase in liver MDA levels, which is representative of oxidative stress and a marked decrease in antioxidants; GSH, GST, SOD and CAT. Infected groups treated with PZQ or AL or CU showed improvement in these parameters compared to infected untreated mice.
The results obtained using Allicina (AL) from Allum sativum and Curcumina (CU) from Curcuma longa, which are natural compounds based on the phenolic component, give promising results for the purpose of replacing the synthetic drug (PZO) praziquantel.
The obtained results are a solid basis for further research on natural remedies.
We improved the English in the text in a red color
Thanks very much
The third referee
In this manuscript, the authors focused on evaluating the efficacy of allicin and curcumin as anti-schistosomal drugs and their biochemical effects in normal and Schistosoma mansoni-infected mice.
The approach used for these techniques is scientifically sound and the results correctly presented. The theoretical research involved in this manuscript is of great significance. But the author did not elaborate on the important significance. It is suggested that the significance of the manuscript research should be emphasized and the rigor of the manuscript logic should be increased. However, the English in the manuscript is understandable but can be improved. The authors should revise the manuscript for language. Moreover, the authors should supplement some more meaningful experiments and refer to more literatures to show the importance of this work. Overall, I don't think the manuscript is suitable for publication.
Specific comments
- The introduction part lacks logic and the research significance of this manuscript is not highlighted. It is recommended to reorganize.
The aim: Consequently, the present work was aimed to evaluate the antischistosomal activity of the active ingredients of two essential traditional plants in infected mice and compared with PZQ efficacy. Parasitological, biochemical and molecular parameters were used to qualify the efficiency of Al and CU as well as grasp the link between these different parameters. Lines 68-72.
- Standard substances such as allicin and curcumin in the manuscript should be marked with purity.
The purity of the standard substances such as allicin and curcumin is added in section 4.1 of chemicals
- Payattention to the format of references, and there are too many references in this manuscript. Too many references can not show the level of the manuscript well. It is recommended to cite references in recent years.
We change the old reference with the recent one and try to reduce the number of references but we found all important. We correct the formate of the references
Aula, O.P., McManus, D.P., Jones, M.K. and Gordon, C.A., 2021. Schistosomiasis with a Focus on Africa. Tropical Med Infect Dise, 6(3): p.109.
Tamarozzi, F., Fittipaldo, V.A., Orth, H.M., Richter, J., Buonfrate, D., Riccardi, N. and Gobbi, F.G., 2021. Diagnosis and clinical management of hepatosplenic schistosomiasis: A scoping review of the literature. PLoS Neglec Trop Disea, 15(3): p.e0009191.
Dejon-Agobé, J.C., Adegnika, A.A. and Grobusch, M.P., 2021. Haematological changes in Schistosoma haematobium infections in school children in Gabon. Infection, pp.1-7.
- I think the results part is just a simple listing, and does not deeply link the experiment results to explain
We discuss every point in the result
- The presentation of the experimen results is too simple and not very attractive.
We try to make it simple so any one can understand the points.
- The definition of the Figure horizontal and vertical coordinates is not high, it is recommended to adjust.